A real-time approach of diagnosing rice leaf disease using deep learning-based faster R-CNN framework

Bari Bifta Sama 1 biftasama.eee@gmail.com
Islam Md Nahidul 1
Rashid Mamunur 1
Hasan Md Jahid 2
Razman Mohd Azraai Mohd 2
Musa Rabiu Muazu 3
Ab Nasir Ahmad Fakhri 2 4
http://orcid.org/0000-0002-3094-5596 P.P. Abdul Majeed Anwar 2 4 amajeed@ump.edu.my
1 Faculty of Electrical & Electronics Engineering Technology, Universiti Malaysia Pahang , Pekan, Pahang , Malaysia
2 Innovative Manufacturing, Mechatronics and Sports Laboratory, Faculty of Manufacturing and Mechatronic Engineering Technology, Universiti Malaysia Pahang , Pekan, Pahang , Malaysia
3 Centre for Fundamental and Continuing Education, Universiti Malaysia Terengganu , Kuala Nerus, Terengganu , Malaysia
4 Centre for Software Development & Integrated Computing, Universiti Malaysia Pahang, Pahang Darul Makmur , Pekan , Malaysia
Damaševičius Robertas
Electronic publication date: 2021 Apr 7
Publication date: 2021
Volume: 7
Electronic Location ID: e432
Received 2020 Nov 23; Accepted 2021 Feb 17
Copyright: © 2021 Bari et al.
Copyright year: 2021
Copyright holder: Bari et al.
License: This is an open access article distributed under the terms of the Creative Commons Attribution License, which permits unrestricted use, distribution, reproduction and adaptation in any medium and for any purpose provided that it is properly attributed. For attribution, the original author(s), title, publication source (PeerJ Computer Science) and either DOI or URL of the article must be cited.
License URL: https://creativecommons.org/licenses/by/4.0/

Keywords: Faster R-CNN, Object detection, Rice reaf disease detection, Image processing, Deep learning

Funding: Universiti Malaysia Pahang RDU200332 This work was supported by the Universiti Malaysia Pahang through the research grant RDU200332. The funders had no role in study design, data collection and analysis, decision to publish, or preparation of the manuscript.

==============================
The rice leaves related diseases often pose threats to the sustainable production of rice affecting many farmers around the world. Early diagnosis and appropriate remedy of the rice leaf infection is crucial in facilitating healthy growth of the rice plants to ensure adequate supply and food security to the rapidly increasing population. Therefore, machine-driven disease diagnosis systems could mitigate the limitations of the conventional methods for leaf disease diagnosis techniques that is often time-consuming, inaccurate, and expensive. Nowadays, computer-assisted rice leaf disease diagnosis systems are becoming very popular. However, several limitations ranging from strong image backgrounds, vague symptoms’ edge, dissimilarity in the image capturing weather, lack of real field rice leaf image data, variation in symptoms from the same infection, multiple infections producing similar symptoms, and lack of efficient real-time system mar the efficacy of the system and its usage. To mitigate the aforesaid problems, a faster region-based convolutional neural network (Faster R-CNN) was employed for the real-time detection of rice leaf diseases in the present research. The Faster R-CNN algorithm introduces advanced RPN architecture that addresses the object location very precisely to generate candidate regions. The robustness of the Faster R-CNN model is enhanced by training the model with publicly available online and own real-field rice leaf datasets. The proposed deep-learning-based approach was observed to be effective in the automatic diagnosis of three discriminative rice leaf diseases including rice blast, brown spot, and hispa with an accuracy of 98.09%, 98.85%, and 99.17% respectively. Moreover, the model was able to identify a healthy rice leaf with an accuracy of 99.25%. The results obtained herein demonstrated that the Faster R-CNN model offers a high-performing rice leaf infection identification system that could diagnose the most common rice diseases more precisely in real-time.

Introduction

Plant disease has become a serious threat towards the production as well as the provision of food security all over the world. For instance, it was reported that more than 800 million people globally lack sufficient food, about 10 percent of the world’s food supply is lost due to plant disease which significantly affects over 1.3 billion people who survive on less than $1 per day (Strange & Scott, 2005; Christou & Twyman, 2004). It is worth noting that plant diseases lead to 10–16 percent annual losses by costing an estimated US$ 220 billion in global crop harvests (Society for General Microbiology, 2011). These statistics portrayed the lingering food scarcity as a result of damage to food production induced by plant diseases that have become a global issue which should not be overlooked by plant pathologists (Strange & Scott, 2005; Ng, 2016). Therefore, to ensure an adequate supply of food to the rapidly increasing population, agricultural production must be raised by up to 70 percent. Nonetheless, several factors militate against the provision as well as the supply of the food to satisfy the need of the teeming population globally.

Rice is amongst the widely consumed food in the world with the total consumption of 493.13 million metric tons in 2019–2020 and 486.62 in the year 2018–2019 (Shahbandeh, 2021). This has shown an increase in the consumption of rice when compared with the metric tons consumed across the years. It is expected that the increase in the consumption of rice tallies with production rates. However, the absence or lack of proper monitoring of farmland often resulted in the destruction of a large amount of rice emanating from diseases related problems. Several diseases frequently occur in the cultivation of rice which is the key reason for major economic losses. In addition, the abundant utilization of chemicals, for example, bactericides, fungicides, and nematicides have produced adverse effects in the agro-ecosystem to combat plant diseases (Nagaraju & Chawla, 2020).

Disease prediction and forecasting of rice leaves are essential in order to preserve the quantity and quality of rice production since detection at the initial stage of the disease are useful in ensuring that timely intervention could be provided to convert the growth of the disease to facilitate the healthy growth of the plant for increasing the production as well as the supply of the rice (Barbedo, 2016). Generally, the rice diseases are sheath blight, bacterial blight, rice blast and symptoms characterized by texture, the color and the shape, that are typical of rapid occurrence and easy infection (Zarbafi & Ham, 2019; Han et al., 2014; Sibin, Duanpin & Xing-hua, 2010). The artificial identification, querying rice diseases maps, and automated detection are currently considered as the procedure of rice disease detection.

The conventional means of rice diseases identification are often carried out manually and has shown to be unreliable, expensive as well as time-consuming. The mapping technique of rice disease detection is relatively clear and easier to perform; however, it is possible to misinterpret some extremely similar diseases that have negative impacts on the growth of the rice. The latest computer-based identification system is yet to be broadly implemented due to the large environmental effect, slow detection speed, and low accuracy. Hence, developing a rice disease detection technique that could provide quick and accurate decisions on rice diseases is of great significance. Rice diseases are localized in leaves, which can be directed by leaf diagnosis to guide growers on whether the crops should be sprayed. It is worth noting that to date, a substantial progress has been made in the detection of plant diseases through the leaf features (Phadikar & Sil, 2008; Hwang & Haddad, 1995; Pantazi, Moshou & Tamouridou, 2019; Rahnemoonfar & Sheppard, 2017; Zhang et al., 2019).

Many researchers have worked on the automatic diagnosis of rice diseases through conventional means such as pattern recognition techniques (Phadikar & Sil, 2008; Rahman et al., 2020), support vector machine (Phadikar, Sil & Das, 2012; Prajapati, Shah & Dabhi, 2017), digital image processing techniques (Arnal Barbedo, 2013; Zhou et al., 2013; Sanyal et al., 2008; Sanyal & Patel, 2008) and computer vision (Asfarian et al., 2014) for enhancing the accuracy and rapidity of diagnosing the results. In an earlier study, Phadikar & Sil (2008) proposed a rice disease identification approach where the diseased rice images were classified utilizing Self Organizing Map (SOM) (via neural network) in which the train images were obtained by extracting the features of the infected parts of the leave while four different types of images were applied for testing purposes. A somewhat satisfactory classification results were reported. In a different study, Phadikar, Sil & Das (2012) proposed an automated approach to classify the rice plant diseases, namely leaf brown spot and the leaf blast diseases based on the morphological changes. A total of 1,000 spot images captured by Nikon COOLPIX P4 digital camera from a rice field were used. The results obtained were 79.5% and 68.1% accuracies from the Bayes’ and SVM classifiers, respectively.

Support Vector Machine (SVM) technique was also utilized by Prajapati, Shah & Dabhi (2017) for multi-class classification to identify three types of rice diseases (bacterial leaf blight, brown spot, and leaf smut). The images of infected rice plants were captured using a digital camera from a rice field and obtained 93.33% accuracy on training dataset and 73.33% accuracy on the test dataset. Zhou et al. (2013) investigated a technique to evaluate the degree of hopper infestation in rice crops where a fuzzy C-means algorithm was used to classify the regions into one of four classes: no infestation, mild infestation, moderate infestation and severe infestation. Their study illustrated that the accuracy reached 87% to differentiate cases in which rice plant-hopper infestation had occurred or not whilst the accuracy to differentiate four groups was 63.5%. Sanyal et al. (2008) proposed an approach for detecting and classifying six types of mineral deficiencies in rice crops where each kind of feature (texture and color) was submitted to its own specific multi-layer perceptron (MLP) based neural network. Both networks consist of one hidden layer with a different number (40 for texture and 70 for color) of neurons in the hidden layer where 88.56% of the pixels were correctly classified. Similarly, the same authors proposed another similar work (Sanyal & Patel, 2008) where two kinds of diseases (blast and brown spots) that affect rice crops were successfully identified. Asfarian et al. (2014) developed a new approach of texture analysis to identify four rice diseases (bacterial leaf blight, blast, brown spot and tungro virus) using fractal Fourier. In their proposed study, the image of the rice leaf was converted to CIELab color space and the system was able to achieve an of accuracy 92.5%.

The feature extraction from diseased and unaffected leaf images, the gray level co-occurrence matrix (GLCM) and the color moment of the leaf lesion region were implemented by Ghyar & Birajdar (2018) to create a 21-D feature vector and related features. The redundant features were eliminated with the genetic algorithm-based feature selection method to generate 14-D feature vectors to minimize complexity. The technique has shown a promising result; however, to improve its detection accuracy there is need for more optimization procedure to take place. The rice disease from the brown spot and blast diseases was described utilizing the color texture of rice leaf photos by Sanyal & Patel (2008). However, the technological standard of identification of rice diseases needs to be strengthened. In Phadikar & Sil (2008), the entropy-based bipolar threshold technique was employed for segmentation of the image after improving its brightness and contrast. The author sought to integrate the image processing and soft computing technique for the detection of rice plant attacked by several types of diseases. The idea behind the technique was robust when utilized effectively. However, the average accuracy of identification on the four datasets was 82 percent which indicates that more enhancement is still required. The image processing and machine learning methods were utilized to non-destructively screen seedlings with rickets by Chung et al. (2016). Moreover, genetic algorithms were employed to develop SVM classifiers in order to optimize feature selection and model parameters for differentiating healthy seedlings and infected ones. The overall accuracy achieved in their study was 87.9 percent. However, since various diseases may have several symptoms, this approach should be tested if it is needed to use in other diseases, suggesting that this procedure has some limitations.

Nonetheless, it is worth noting that researchers have also begun to move away from such techniques to deep learning models in an effort to detect diseases in various plants (DeChant et al., 2017; Zhang et al., 2018b; Zhang & Zhang, 2010; Liu et al., 2017). The Convolutional Neural Networks (CNN) is a deep learning method that has become one of the best image classification technique which has already acquired great success (Xu et al., 2017; Zhao & Jia, 2016; Sainath et al., 2015; Ribeiro et al., 2016; Ciresan et al., 2011; Kawasaki et al., 2015). A rice disease identification system based on a deep convolutional neural network was reported by Lu et al. (2017b). It was observed that the average identification rate was 95.48 percent for 10 common rice diseases using the 10-fold cross-validation scheme. Zhou et al. (2019) suggested Faster R-CNN approach, which seems to be ideal for the detection of rice diseases due to its good speed and high accuracy. Another method suggested by Ren et al. (2017) was capable of detecting plant diseases as well as enhancing the accuracy using Faster R-CNN. However, it is required to reduce the time for disease identification in order to allow it to be suitable for monitoring large-scale cultivation.

These advanced techniques are used not only for the rice diseases diagnosis but also for some other crops including wheat (Lu et al., 2017a; Khairnar & Dagade, 2014), maize (Zhang & Yang, 2014), pumpkin (Zhang et al., 2018a), cotton (He et al., 2013) and tomato (Wang et al., 2019), amongst others. DeChant et al. (2017) proposed a three-stage architecture consisting of multiple convolutional neural networks (CNNs) where the stage-one model is trained on full-scaled images by dividing a single image into several smaller images. On the other hand, two improved deep convolution neural network models (GoogLeNet and Cifar10) were utilized by Zhang et al. (2018b) to improve the recognition accuracy of the maize leaf diseases and enhance the traditional identification techniques that often require long convergence times and large numbers of model parameters. It was shown from the study that an average accuracy of 98.9% and 98.8%, respectively are attainable. Liu et al. (2017) developed an apple leaf disease identification technique that includes of generating sufficient pathological images and designing a novel architecture of a deep convolutional neural network based on AlexNet that was able to achieve an overall accuracy of 97.62%. The CNN approach has also been applied by Martin & Rybicki (1998) to classify the Helminthosporium leaf spot of wheat, and an accuracy of 91.43% and standard error of 0.83% were recorded.

Fuentes et al. (2017) proposed a deep-learning-based approach using three architectures, namely, Faster Region-based Convolutional Neural Network (Faster R-CNN), Region-based Fully Convolutional Network (R-FCN), and Single Shot Multibox Detector (SSD) that can effectively recognize nine different types of diseases and pests in tomato plants. In a recent study, Rahman et al. (2020) developed a CNN approach for detecting diseases and pests (five classes of diseases, three classes of pests and one class of healthy plant and others) from rice plant images. A total number of 1,426 images were collected that were captured using four different types of cameras and the system achieved a mean validation accuracy of 94.33 %. Kawasaki et al. (2015) suggested a method to identify cucumber leaf disease based on CNNs by achieving 94.9 percent accuracy in distinguishing between melon yellow spot virus, zucchini yellow mosaic virus, and non-diseased type virus. A new stacked CNN architecture is suggested by Rahman et al. (2020) which uses two-stage training to substantially reduce the model size while retaining high classification accuracy. It was found that the test accuracy was able to achieve 95 percent using stacked CNN compared to VGG16, while the model size was reduced by 98 percent.

The development of a technique for automatic identification of rice leaf disease is hitherto faced many challenges. It is noted that the diagnosis, as well as detection, involves processes that could render the specific area in which the symptoms manifest within the rice plant very difficult to segment correctly. The capture conditions are hard to handle, which can make it harder to predict images and make detection of the disease more difficult. Moreover, the symptoms caused in different diseases can be identical visually, and the approaches of discrimination could be based on very tiny variations. Another very common issue is the discrepancies in the distribution of the data features to train the model as well as the data that could be used to validate the model. This situation creates overfitting problem. This is very important when plant diseases are automatically detected because the symptoms can differ from the geographical position and fall into the overfitting problem. It has also been observed that many of the suggested rice leaf disease diagnostic architectures are off-line, and only a few experiments have been carried out in real-time. Usually, the image resolution is enhanced in real-time by which the computational complexity should also be enhanced. In addition, the difficulty of real-time operations increases with a large variety of disease features, complex backgrounds and obscure boundaries of the disease symptoms. In order to address these challenges, the current study endeavors to employ the latest deep learning approach based on Faster R-CNN to conduct real-time detection of rice leaf diseases. The present investigation is sought to mitigate the lingering problems in the process of developing a system of diagnosing rice disease. The key contributions of the research are summed up as follows:Disease spot identification is considered as the basis of recognition for rice leaf disease, as such the accuracy of spot identification directly impacts on the accuracy of recognition of rice leaf disease. Hence, when choosing the target detection algorithm, recognition accuracy should be employed as the key indicator. YOLO, SSD and Faster R-CNN are the mainstream algorithms for the detection of the deep learning target. Among them, the Faster R-CNN algorithm creatively proposes the RPN structure to generate candidate regions, making the target positioning very precise. In addition, Faster R-CNN also has strong advantages in detection accuracy compared to YOLO and SSD. The proposed study employed Faster R-CNN as the key research algorithm due to its efficacy in detecting the spot of the disease reliably.

The data set for rice leaf disease is designed to provide a significant guarantee of the proposed model’s generalization capability. Here, diseased rice leaf images with standardized and complex backgrounds are captured both in the lab and in real field conditions to improve the robustness of the Faster R-CNN model. In addition, natural-diseased rice leaf images are processed to produce sufficient training images through data augmentation technology in order to solve the complexity of insufficient diseased rice leaf images and to avoid overfitting of Faster R-CNN models in the training phase.

A Faster R-CNN network is employed for the real-time detection of rice leaf diseases. With the proposed deep-learning method, the discriminatory features of diseased rice images will automatically be classified, and the three major types of rice leaf diseases are recognized with high accuracy. Furthermore, the proposed method could manage all the rice leaf images collected from the rice farmland in real conditions.

The present manuscript is structured as follows: rice leaf diseases dataset (RLDD) generation techniques are implemented in the Materials and Methodology section. A detailed description of the development of the model for the detection for the rice leaf diseases is described in this section. Experimental outcomes to determine the accuracy of the proposed solution are described in the Results section, and the Discussion section exhibits a discussion on the comparison of the proposed model with other related studies along with limitations with prospective solutions for rice leaf disease detection approaches, followed by the Conclusion, which draws the outcome of the present study.

Materials and methodology

Figure 1 shows the comprehensive procedure of real-time identification. First of all, RLDD is constructed using a combination of an online database and an own dataset that was collected in this experiment. The online database is freely accessible. The own dataset was created by capturing diseased rice leaf images in the laboratory which were collected by the authors from actual rice fields. The original RLDD is then annotated manually and expanded through the several data augmentation procedures. The entire dataset is subsequently split into two groups: training dataset and testing dataset. To train the Faster R-CNN model, training dataset is employed whereas testing dataset is utilized for performance assessment. The detection outcomes consist of the classes as well as the locations of the identified rice leaf diseases.

Figure 1 Complete architecture of the proposed study.

Data collection

Due to the absence of adequate data for real-time rice leaf disease, some of our authors and material resources were committed at the start of our study to collect diseased rice leaves. The patterns of diseases of rice leaves are varied over the season and other factors including moisture, temperature, different insects and illuminance. For example, most conspicuous symptoms of brown leaf spot disease occur on leaves and glumes of maturing plants. In real-time operation, the data collection process is very important since the inappropriate information in a dataset may hamper the experimental result. Hence, during the data collection process, the standard rule should be introduced and maintained.

In this study, the rice images have been captured from the rice farmland and a different condition of the leaves were collected and brought to the lab. The rice leaf datasets were also collected from Do (2020) due to the lack of suitable and different conditions data from real field. It also helps to check the validation of the proposed model. Then, the entire datasets were merged to train the model and each data has been checked individually to avoid misclassification. The rice leaf infected image database consists of healthy leaf and three diseases including rice blast, brown spot, and hispa. To enhance the robustness of the proposed system, our own captured rice leaf image is combined with a publicly available online database.

From the dataset, 600 images of rice blast, 650 images of brown spot, 500 images of hispa and 650 images of healthy rice leaf have been collected. A total number of 2,400 images were collected. The total number of images collected from each database (Kaggle and own dataset) are summarized in Table 1.

Table 1 Total number of images collected from each database.

Leaf condition	Kaggle dataset (publicly available)	Own dataset (on-field dataset)	
Rice blast	500	100	
Brown spot	500	150	
Hispa	500	–	
Healthy	500	150	
Total	2,000	400	
2,400	

Data augmentation

Data augmentation is the process of broadening the dataset to enhance the model’s performance by generating different forms of images. It also serves useful in mitigating the overfitting problem in the model during the training stage. The overfitting problem occurs when there is the presence of random noise or errors, rather than the underlying relationship. With the help of data augmentation, more image was generated from each image to train the model since some irrelevant patterns may occur during the training process of the model. For data augmentation operations, several techniques were used namely, rotation transformations, horizontal and vertical flips, as well as intensity disturbance which includes disturbances of brightness. A Gaussian noise processing scheme is employed in which the natural sources like thermal are responsible for the Gaussian noise. It is worth noting that in digital images, Gaussian noise interrupts the gray values. To train the model with training data set, Gaussian noise images were used for better results. With the above approaches, 7 new images are generated from each image as shown in Fig. 2. Finally, the dataset containing 16,800 images were created using the data augmentation technique.

Figure 2 Data augmentation of rice leaf disease images: (A) original image (B) image rotated by 180 degree (C) high brightness (D) Gaussian noise (E) horizontal flip (F) low brightness (G) vertical flip.

Image annotation

Image annotation plays a key role in labeling the positions and classes of object spots in the disease and healthy images for multiclass object detection. In computer vision, Pascal VOC is the method which stores annotation in the XML file and the separate annotation files are saved for each image. The LabelIMG is the graphical image tool used for this process in VOC format which is developed in python. The Pascal VOC provides standardized image data sets for object detection. We constructed a file for every image of the dataset in the Pascal VOC. The XML file created includes information such as the bounding box coordinate values and the disease classes. For training purposes, 400 images were annotated for each class (rice blast, hispa, brown spots, healthy) from the dataset and the rest of the images for testing our model performance. Although the whole process is very challenging owing to the fact that the disease area seems to be tiny and difficult to detect a times, nonetheless, it is worth highlighting that a high detection performance in our model was observed. The Fig. 3 shows the annotated images of the brown spots.

Figure 3 The image annotation outcome in XML file.

Figure 3 shows that the image contains the object details. The object tag and its content are replicated when images have several annotations. The object tag components are name, pose, truncated, difficult and bound box. These are the names of the objects that are to be detected. Truncated states the bounding box that the object specifies does not fit the entire extent of the object. If an object is partly visible in the image, the truncated is set to 1. Otherwise, the object truncated is set to 0, if completely visible. Difficult: When the object is seen as difficult to identify, an object is identified as difficult. If the object is difficult to recognize, then difficult is set to 1, else is set to 0. The axis-aligned rectangle in the bounding box indicates the size of the object is visible in the image. This technique contributed to understanding the specifics of the two common computer vision data formats.

Model architecture with Faster R-CNN

A new effort Faster R-CNN (Ren et al., 2017) was launched in 2015 by the team in the target detection community with Ross Girshick after R-CNN (Girshick et al., 2014) and Fast R-CNN (Girshick, 2015) were launched. The R-CNN approach is very important for understanding proposal regions since the proposal regions are classified into object categories or background by training of CNNs end-to-end in R-CNN technique (Girshick et al., 2014). Basically, R-CNN works as a network classifier. The accuracy of the model is based on the performance of the region proposal module (Ren et al., 2017). Faster R-CNN does not need a fixed size to detect rice diseases image. As an image input, the length and width must be limited to a certain degree, thereby preventing distortion. The detection speed is significantly increased after the enhancement of the Regional Proposal Network (RPN). An integration of the region proposal algorithm into a CNN model will lead to a faster speedup implementation (Ren et al., 2017). This approach is mainly conducted by Faster R-CNN in order to build a single and unified model that is consisted of region proposal network (RPN) and fast R-CNN with shared convolutional feature layers. Instead of selective search in Fast R-CNN, Faster R-CNN may be simply regarded as a model of “the regional generation network + Fast R-CNN” that employs the RPN which is a recommendation algorithm for this propose. The convolution layer/full connection layer processing takes place on the feature map, then a position regression and classification is applied to the detected target. The recommendation of the region is utilized to secure a better location of the disease. Fast R-CNN refers to the detailed measurement of the frame position and object categories in the frame. The following Box 1 exhibits the steps of Faster R-CNN are used to build the model for rice leaf disease detection.

Box 1 Steps of the Faster R-CNN technique.

The Faster R-CNN technique:	
Step 1:	To acquire a feature map, the entire image of rice diseases is fed into CNN	
Step 2:	To gain the feature information of the candidate frame and the convolution feature is then fed into the RPN	
Step 3:	To recognize whether the features of rice diseases from the candidate box belongs to a specific disease category and then classify	
Step 4:	To adjust the disease location again by a regression device for the candidate frame belonging to a specific disease feature	

RPNs for candidate regions

The main concept of RPN is to produce regions utilizing CNN that are explicitly suggested. The shared convolution network is fed by the rice leaf diseased images where feature map is achieved that is used as RPN input. The convolutional feature map points are the original image positions (Girshick et al., 2016). The components on every map are nine anchor boxes of several sizes. There are two convolutional networks in the RPN. One is a convolution technique of 18-dimensional using a 1 × 1 convolution kernel to decide about a foreground image which belongs to the anchor box or not. The other is a 1 × 1 volume with which another convolution structure is passed. To achieve the relative position coordinates dx(A), dy(A) and dw(A) of the bounding box in the case of Ground Truth, a 36-dimensional convolution mechanism is conducted by the accumulative kernel. The original image is mapped with every point on the feature map in which “anchor point” is described by each pixel (Ramaswamy et al., 2014). Each anchor point is utilized to be positioned of multiple anchors of different sizes. The mostly utilized 3 varied aspect ratios are 2:1, 1:1 and 1:2 for popular scales 5122, 1282 and 2562 respectively. The new rectangular position achieved by the anchor is modified at first by the adjustment parameters in the proposal laying of the RPN. The position vector of the lower-left edge and top edge within each target area are considered as the outcomes for the target areas of the earliest photo. This is how the RPN measures are as follows (Box 2):Feature extraction

The processed RPN image is sent to the layer of RoI Pooling, which pools the areas of rice diseases. By further enhancing the SPP-Net algorithm, the Faster R-CNN algorithm suggests a region of interest (RoI Pooling). The RoI Pooling layer enables a number of dimensions to be transformed into a fixed size in line with the needs of the next fully connected network. Every rice disease candidate’s area is equally divided by the ROI pooling layer in M × N blocks and performs maximum pooling per block (Chang et al., 2019). On the rice disease map, disease candidates of different sizes are converted into standardized data and forwarded to the following layer. Although the size of the input image and the feature mapping is different, a feature representation of a fixed dimension can be extracted for each area by applying the ROI pooling layer to define the disease classification later.

Box 2 Steps of the RPN for candidate regions.

RPN steps for candidate regions:	
Step 1:	To slide a window on the map of rice disease	
Step 2:	To classify the leaf infections and revert back the location of the frame, a neural network is formed.	
Step 3:	To provide approximate distribution details of leaf infection according to the position of the sliding window	
Step 4:	To achieve a better location of leaf infection with the box’s regression	

Classification, regression and location refinement

The diseases are classified, and the position is refined by taking into account the pictures of rice diseases. The classification steps shall be: first, the classification of objects or non-objects for each of the two regions corresponding to the Anchor Box, k then models of regression (both equal to a different Anchor Box). The formula (Eq. (1)) for the complete estimation of the classification layer is as follows:

(1) (x1x2x3)(w11w12w21w22w31w32)+(b1b2)=(y1y2)

The rice disease location is determined by the size of the overlap region. The inaccuracy of the candidate’s frame and the slight overlap are often the main reasons for unreliable test results. Therefore, a judicious identification of the location is non-trivial towards attaining encouraging results. The eigenvectors achieved in the classification are determined by a complete connection and Softmax, and a species is generated with a probability of a certain rice disease species. The anchor box regression is used to compensate the region from its actual GT position, hence closer to the real position of the rice disease detection frame.The training processes and loss function

The Caffe deep learning approach is used to carry out the experiment. The training set of rice diseases was sent randomly to the neural network for training. The model was tested, and the test results were analyzed after the completion of the training process. The following Box 3 reflects the phases for the Faster R-CNN training model:

Box 3 Phases of the training processes (Faster R-CNN training model).

Training processes: Different Phases of Faster R-CNN training model:	
Phase 1:	After initializing the RPN structure with the pre-trained framework, the RPN is trained. The model’s distinctive value and RPN are revised when the training is finished	
Phase 2:	The Faster R-CNN architecture is formed. Subsequently the proposal is calculated by utilizing the trained RPN and then the proposal is sent to the Faster R-CNN network. Following this, the network is trained. Then the model and the uniqueness of the Faster R-CNN is updated through the training process	
Phase 3:	The RPN network is initialized by employing the model that was formed in the Phase 2. Then a second training is carried out on the RPN network. The RPN’s distinctive value is altered at the time of the training procedure while the model parameters remain unchanged.	
Phase 4:	The model variables stated in Phase 3 are kept unaltered. The Faster R-CNN architecture is formed and trained the network for the 2nd attempt to optimize the specifications	

Faster R-CNN is optimized for a multi-task loss function (Girshick, 2015). The multi-task loss function combines the losses of classification and bounding box regression. For training RPNs, a binary class label (of being an object or not) has been assigned to each anchor. Equation (2) represented a loss function for an image following the multi-task loss in Fast R-CNN (Alamsyah & Fachrurrozi, 2019; Ren et al., 2017).

(2) L({pi},{ti})=1Ncls∑i⁡Lcls(pi,pi∗)+λ1Nreg∑i⁡pi∗Lreg(ti,ti∗)

where, i is the index of an anchor, pi is the predicted probability of anchor and pi∗ is the ground-truth label. ti is a vector that represents the 4 parameterized coordinates of the predicted bounding box and ti∗ is the ground-truth box associated with a positive anchor. The classification and regression loss are represented by Lcls and Lreg respectively. λ is a balancing parameter. The Lcls and Lreg normalized by Ncls and Nreg respectively that are weighted by λ. Thus, for the regression loss,

(3) Lreg(ti,ti∗)=R(ti−ti∗)

where, R is represented as a robust loss function.

The complete architecture of a Faster R-CNN is presented in Fig. 4.

Figure 4 Architecture of Faster R-CNN.

Figure 4 illustrates an entire framework for object detection which is a single and unified network. At first, the feature maps are received from CNN by the Faster RCNN. After that, it passes the collected features to the Region Proposal Network (RPN). Various image sizes can be fed as input to the region proposal network (RPN). The outputs are comprised of a series of rectangular object proposals. RPN is inserted next to the last convolution layer of CNN. A small network is slided over the convolutional feature map output for creating region proposal network by the last shared convolutional layer in CNN. A n × n Spatial Window of the input convolutional feature map is the input for this small network. At the position of each sliding-window, multiple region proposals are predicted simultaneously. RPN transmits the last layer of CNN (sliding window) to a lower dimension into feature map. The proposal from RPN are fed to ROI pooling layer. The fixed-size feature maps are generated from different sizes of inputs by ROI pooling layers. The output fixed dimension of the ROI pooling depends on the parameters of the layer. Finally, this feature is used to fed into two fully connected layer, namely box-classification layer (Classifier) and box-regression layer (regressor). A refined bounding box is utilized as a regressor whereas the objects are classified by the classifier. The following equaltions (Eqs. (4)–(11)) (Girshick et al., 2014; Ren et al., 2017) are used for bounding box regression.

(4) tx=(x−xa)/wa,

(5) tw=log(w/wa),

(6) tx∗=(x∗−xa)/wa,

(7) tw∗=log(w∗/wa),

(8) ty=(y−ya)/ha,

(9) th=log(h/ha),

(10) ty∗=(y∗−ya)/ha,

(11) th∗=log(h∗/ha),

where the center coordinates, width and height of the box are represented by x, y, w, and h. The variables x, xa, and x∗ (similarly applicable for y, w and h also) are for the predicted box, anchor box, and groundtruth box respectively.

Results

Feature visualization process

Owing to the limited explanatory nature of CNN, visualization methods are often used to further understand the CNN features maps in order to decide how CNN’s can learn features of the different class evaluated. This experiment is carried out to comprehend better the variations between the feature maps extracted from different diseased rice leaf images. The visualization outcomes are shown in Fig. 5, which suggest that all the disease spots are clearly identified from the background images. Therefore, the proposed model demonstrates excellent performance in the discrimination of rice leaf diseases.

Figure 5 Activation visualization results: (A) Rice Blast (B) Brown Spot (C) Healthy (D) Hispa.

The visualization outcome for the healthy leaf is shown in Fig. 5C. Figure 5D indicates Hispa that is commonly occurred in a very small region, and the boundaries are not explicit. For brown spot, the spots are divided into two laps, as shown in Fig. 5B. In addition, Fig. 5A explores the Rice Blast, which is almost similar to Brown Spot as shown in Fig. 5B. They can still be identified according to their minute differences. This experiment demonstrates the strong performance in the design of the proposed model for disease detection, and it clarifies how the CNNs can differentiate between classes by visualizing the features of different rice leaf diseases.

Detection visualization and failure analysis

The outcomes for the identification of rice leaf disease are shown in Figs. 6 and 7. The proposed approach can identify both a single object and multiple objects of a single class, also multiple objects of multiple classes. The proposed method therefore demonstrates high detection performance in both single and multi-class assessments. Although the model is excellent in terms of accuracy, there are inevitable detection failures which occur when the spot region of the leaf is too small. A rice leaf disease example is illustrated in Figs. 6F and 6H containing two leaf disease types in a single class. The proposed model is able to detect rice blast and hispa diseases of this class, but the small portion of hispa disease is not detected successfully. The model detects hispa and a healthy portion of the leaf successfully, as shown in Fig. 6H. On the other hand, from the Fig. 6F, it is evident that the model can detect the multi-class disease (rice blast and hispa) efficiently; however, it fails to detect the very tiny portion of hispa. The reduction in the detection accuracy is attributed to the similar characteristics of the diseases as shown in Fig. 6E. Owing to the similar characteristics of brown-spot and rice-blast, the developed model was confused in some cases. Environmental variables including complex background, blurriness and lighting also influence the accuracy of identification. Furthermore, one of the factors contributing to increase the detection failure is the small size of the lesion. Hence, it will be difficult to extract and detect the feature if only a small part of the image is taken by the leaf or the diseased region. Despite of all the limitations, in most of the cases, the proposed model has the ability to detect the leaf spot as shown in Figs. 6A, 6B, 6C, 6D and 6G. The detection ability of the leaf spot in a real rice field is presented in Fig. 7.

Figure 6 Types of detection results (Images collected from Online and captured in the lab).

(A) Brown Spot, (B) Healthy, (C) Hispa, (D) Rice Blast, (E) Brown Spot and Rice Blast, (F) Hispa and Rice blast, (G) Rice Blast, (H) Hispa and Healthy.

Figure 7 Types of detection results (real field image).

(A) Rice Blast. (B) Healthy.

Comparison of pre-network recognition accuracy

Object detection algorithms like Single Shot Detector (SSD), Deconvolutional Single Shot Detector (DSSD) and Rainbow Single Shot Detector (R-SSD) essentially consist of two components. The first element is the pre-network model used to extract the basic features. The other is an auxiliary structure that utilizes multi-scale detection of feature maps. Various deep convolution networks including ResNet-101, ResNet-50, and VGGNet-16 (Simonyan & Zisserman, 2015; Liu & Deng, 2016), and MobileNET (Howard et al., 2017) are trained and tested to compare the recognition performances of traditional networks with that of our proposed Faster R-CNN on RLDD. The stochastic gradient descent (SGD) algorithm is employed during training to learn about the neural network weights and biases, which reduces the loss function. A limited number of training sets are selected randomly by the SGD algorithm, known as the batch size. The batch size is set to 1 where the final number of iterations is fixed at 50,965. The learning rate is set at 0.0002, although very small, it contributes towards more reliable results. The momentum, which acts as an additional factor to decide how quickly the SGD algorithm converges to the optimal point, is set at 0.9. The accuracy curve is indicated, as shown in Fig. 8 with the number of training iterations in the X-axis and corresponding Y-axis shows the training accuracy. The comparison of test accuracies of different pre-networks (VGGNet-16, ResNet-50, ResNet-101, MobileNet3 and Faster R-CNN) are defined in terms of accuracy curve, as shown in Fig. 8. The VGGNet-16 networks have higher convergence speed but lower accuracy. On the other hand, from the figure, it is evident that the Faster R-CNN model shows high accuracy on the RLDD as compared to other pre-trained models.

Figure 8 Performance comparison with other pre-trained model.

Confusion matrix

When dealing with multiple classes of similar shape, classifiers may be confused. Infected rice leaf images on different levels or backgrounds can cause high complexity which leads to lower performance for the patterns displayed in the same class. The classification accuracy of a model can be visually tested using a confusion matrix. The entire dataset of our study is split into a training set and a testing set randomly in order to train and test the model. To evaluate the proposed model, the 50% dataset is used to train and the remaining 50% dataset is used to test. Total 8,400 observations are utilized for training the model, whereas another 8,400 observations are utilized for testing the model. Figure 9 displays the final test results confusion matrix. The deeper the color in the visualization results, the greater the model’s accuracy in the respective class. All correct predictions are located diagonally, whilst all wrong predictions are off diagonal. The classification accuracy can be visually assessed based on these findings.

Figure 9 Confusion matrix of the proposed approach.

The study shows that for the above three diseases and healthy leaf. Brown spot and hispa diseases are significantly differentiated from other diseases by their features and by their identification rates with 98.85% and 99.17%, respectively. In the healthy leaf study, the accuracy is achieved by 99.25%. According to the confusion matrix, it is apparent that the detection model is more prone to confusion in distinguishing rice blast and brown spot compared with other classes. Among 2,100 images in the testing set of rice blast spot, 31 images have been detected incorrectly as brown spot. On the other hand, among 2,275 images in the testing set of brown spots 20 images have been detected incorrectly as rice blast spots. This misclassification may be caused by the geometrical feature similarities between the two diseases. However, other classes are well distinguished. The confusion matrix describes the low inaccuracies in the identification of different classes in the present investigation.

Loss analysis of the proposed model

The research seeks to mitigate the loss of function, thereby reducing errors in the model. In doing so, every machine learning algorithm repeats calculations several times, until the loss is plateaued. The learning rate plays a significant role in order to minimize the loss function. In the proposed study, the learning rate is set to 0.0002. TensorBoard is a fantastic tool for viewing these metrics and finding possible problems. TensorBoard frequently upgrades the measurements and provides the outcomes to the user. In this purpose, the model trained with 50,965 iterations with the help of a training dataset. Figure 10 depicts the generated loss analysis by the TensorBoard, indicating that the total loss is withing the vicinity of 0.1.

Figure 10 The classification loss of the proposed system.

Discussion

Comparison of the proposed model with other related studies

The comparison of the proposed model with existing related studies is represented in Table 2.

Table 2 Comparison of the proposed model with other related studies.

Researchers	Methods	Dataset (own or publicly available)	Camera to capture data	Number of observation	Learning rate	Number of iteration	Performance (%)	
Zhou et al. (2019)	FCM-KM and Faster R-CNN fusion	Rice field of the Hunan Rice Research Institute, China	Canon EOS R (pixel: 2,400 * 1,600)	3,010	0.001	15,000	Rice blast: 96.71
Bacterial blight: 97.53
Blight: 98.26	
Sethy et al. (2020)	Faster R-CNN	Farm field	Smartphone camera (48 Megapixel)	50	0.001	5	Initial steps to make a prototype for automatic detection of RFS Rice false smut	
Phadikar, Sil & Das (2012)	Bayes’ and SVM Classifier	Rice field images of East Midnapur, India	Nikon COOLPIX P4 digital camera	1,000	–	–	Normal leaf image: 92
Brown spot image: 96.4
Blast image: 84
Bayes’ classifier: 79.5
SVM: 68.1	
Ramesh & Vydeki (2020)	Optimized Deep Neural Network with Jaya Optimization Algorithm (DNN_JOA)	Farm field	High resolution digital camera	650	–	–	Rice blast: 98.9
Bacterial blight: 95.78
Sheath rot: 92
Brown spot: 94
Normal leaf: 90.57	
Li et al. (2020)	Faster-RCNN	Rice field in Anhui, Jiangxi and Hunan Province, China	Mobile phone camera (iPhone7 & HUAWEI P10) and Sony DSC-QX10 camera	5,320	0.002	50,000	Rice sheath blight: 90.9
Rice stem borer: 71.4
Rice brown spot: 90	
Prajapati, Shah & Dabhi (2017)	SVM	Farm field	NIKON D90 digital SLR (12.3 megapixels)	120	–	–	For SVM:
93.33 (training)
73.33 (testing)
5-fold cross-validation: 83.80
10-fold cross-validation: 88.57	
Narendra Pal Singh Rathore (2020)	CNN	Kaggle dataset	–	1,000	–	–	Prediction accuracy: 99.61 (healthy and leaf blast)	
Rahman et al. (2020)	Simple CNN	Rice fields of Bangladesh Rice Research Institute (BRRI)	Four different types of camera	1,426	0.0001	100	Mean validation accuracy: 94.33	
Proposed Model	Faster-RCNN	Both on field data and Kaggle dataset	Smartphone camera (Xiaomi Redmi 8)	16,800	0.0002	50965	Rice blast: 98.09
Brown spot:98.85
Hispa: 99.17
Healthy rice leaf: 99.25	

Most of the studies listed in Table 2 have used either utilized publicly available dataset or own captured dataset to validate their methods. The models validated with publicly available dataset always do not ensure the stability of the model’s performance in a real-time approach. To address this issue, our proposed method is validated with both publicly available and own dataset. Moreover, the total observation of our proposed study is higher than other studies tabulated in Table 2. Despite these facts, the performance of the proposed model is higher than the other models for rice leaf diseases identification. The detection accuracy achieved by Narendra Pal Singh Rathore (2020) is slightly higher than our proposed method, as their dataset consists of only one type of rice leaf disease (leaf blast), hence the discrepancies are acceptable. Therefore, by considering the strong dataset, real-time disease detection ability and detection accuracy, our proposed method is somewhat superior to that of other related approaches for rice leaf disease identification reported in the literature.

Uncertainties and limitations

Although the proposed model outperforms state-of-art rice leaf diseases detection methods, some of the drawbacks are also identified. Some limitations of this study with the prospective solution to address these challenges are as follows:The network looks at the whole image, and not in just one go but sequentially concentrates on part of the image. Thus, the algorithm requires many passes to extract all objects through a single image which is time-consuming. To address this issue, a network should be recommended which can extract objects of an image in a single pass.

Since several processes have been conducted one after the other, the performance of the further system depends on how the previous system performed. Thus, a model should be trained carefully with appropriate datasets to achieve the desired performance.

The misclassification issues could occur as a result of the geometrical feature similarities between the diseases. To overcome this obstacle, it should be required to train the network with more datasets which have similar geometrical features. It also recommended addressing more efficient deep learning algorithm which can classify the diseases containing small dissimilarities in features.

In a real-time approach, the rice leaves conditions vary with the season having different humidity, temperature, and illuminance. Hence, some overfitting problem may emanate when random noise or errors occurs rather than the underlying relationship, as previously described (Heisel et al., 2017). During the training stage, the overfitting problem is expected to occur randomly due to the lack of proper images with various conditions. To overcome these problems, in this study we have used data augmentation in the process of the training stage of Faster R-CNN. During the training, the proposed model can learn huge irrelevant patterns through the large amounts of images which is generated by the data augmentation process. This phenomenon helps to reduce the overfitting problem and achieve the higher performance. More approaches such as saturation, hue and Generative Adversarial Networks (GANs) (Bowles et al., 2018) can be employed to overcome this issue.

Conclusions

The signs of infection appear in various sections of the plant, and leaves are widely used to diagnose the plant disease. The advanced computer vision technology encourages researchers around the world to carry out extensive experiments on plant disease recognition using leaf image analysis techniques. In the past few years, deep learning methods have notably been utilized to recognize plant leaf infection. This paper proposes a real-time rice leaf disease diagnosis framework based on the Faster R-CNN technique. The rice leaf infected image database consists of healthy leaf and three diseases, including rice blast, brown spot, and hispa. In order to enhance the robustness of the proposed system, our own captured rice leaf image is combined with a publicly available online database. Moreover, we have used several image augmentations schemes to enrich the dataset, which familiarizes the model with the different possible conditions of the image. This strategy also enhances the model’s performance and generalization capability. The obtained results of the proposed study are very encouraging to diagnose healthily and the different types of infected leaves in both laboratory-based images and real-field images. However, an additional study should be carried out to make segmented the infected portions of the leaf image by minimizing the surrounding interference. The existing rice leaf disease diagnosis systems are designed using laboratory-based captured images. Although we have implemented real-time disease recognition architecture using real field rice leaf images, the proposed system is still not fully automated. Therefore, further study should be carried out to implement a dynamic and automatic system to recognize large-scale rice leaf diseases. This system could be made up of a mobile terminal processor and agricultural Internet of Things that may be favorable to modernize the agricultural industry.

Supplemental Information

Supplemental Information 1 Code and instructions.

Click here for additional data file.

Additional Information and Declarations

Competing Interests

Author Contributions

Data Availability

The authors declare that they have no competing interests.

Bifta Sama Bari conceived and designed the experiments, performed the experiments, analyzed the data, performed the computation work, prepared figures and/or tables, and approved the final draft.

Md Nahidul Islam conceived and designed the experiments, performed the experiments, performed the computation work, authored or reviewed drafts of the paper, and approved the final draft.

Mamunur Rashid performed the experiments, analyzed the data, prepared figures and/or tables, authored or reviewed drafts of the paper, and approved the final draft.

Md Jahid Hasan conceived and designed the experiments, analyzed the data, performed the computation work, prepared figures and/or tables, and approved the final draft.

Mohd Azraai Mohd Razman performed the computation work, prepared figures and/or tables, and approved the final draft.

Rabiu Muazu Musa performed the experiments, analyzed the data, authored or reviewed drafts of the paper, and approved the final draft.

Ahmad Fakhri Ab Nasir performed the experiments, analyzed the data, prepared figures and/or tables, and approved the final draft.

Anwar P.P. Abdul Majeed conceived and designed the experiments, performed the experiments, analyzed the data, performed the computation work, prepared figures and/or tables, authored or reviewed drafts of the paper, and approved the final draft.

The following information was supplied regarding data availability:

Code is available in the Supplemental Files.

Data is available at Kaggle:

Do, Huy Minh (2019): Rice Diseases Image Dataset. Kaggle. Dataset. https://www.kaggle.com/minhhuy2810/rice-diseases-image-dataset.

Our data is available at Figshare:

Islam, Md Nahidul (2020): A real-time approach to diagnose the rice leaf disease using deep learning-based faster R-CNN framework. figshare. Dataset. DOI 10.6084/m9.figshare.13270916.v1.

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
