# Peer review of "A real-time approach of diagnosing rice leaf disease using deep learning-based faster R-CNN framework"

_PeerJ Computer Science, doi:10.7717/peerj-cs.432_

## Round 0.1 · original submission · Major Revisions

Recent state-of-the-art works on plant leaf recognition must be discussed. The novelty and contribution of the paper must be explicitly stated. The results achieved should be compared with experimental results from related studies.

Reviewer 1 ·

Basic reporting

No comment

Experimental design

1. Your description of Sub-section "Data collection" is not explanatory enough. Since the experiment was done using two datasets, the first dataset was collected from online database (Kaggle database) and the second database is a primary dataset. Give a detailed summary of the total number of images collected from each database (you can use a Table).

2. For a better view and understanding of the different steps involved in your methodology. Kindly insert all the different stages steps in a textbox with appropriate caption such as:
• The steps of Faster R-CNN:
• RPNs for candidate regions
• Training processes: Faster R-CNN training model:
This would help the reader understand the steps better.

3. Improve discussion of your findings in figures 6 and 7 of Section: Detection and visualization and failure analysis.

4. Figure 9 shows the confusion matrix, it is expected to more detailed as the rate of misclassification of each class is not specified on the figure. Use a more labelled confusion matrix showing the wrong predictions on each class.

Validity of the findings

1. To identify the validity of your findings, it is expected that the results achieved should be compared with experimental results from related studies. Since an online/ publicly available dataset was used (Kaggle database).

2. Highlights what are the limitations or challenges of your model and how it affects the overall performance.

Additional comments

The paper is very interesting to read with the aim of addressing the challenges of real-time rice leaves disease spot identification using Deep Learning-based Faster R-CNN Framework.

Annotated reviews are not available for download in order to protect the identity of reviewers who chose to remain anonymous.

Reviewer 2 ·

Basic reporting

The English language should be improved to ensure that an international audience can clearly understand your text. Some examples where the language could be improved include lines 164-165, 262 the current phrasing makes comprehension difficult.
The summary of the existing technology in the introduction is not comprehensive. I suggest you add a more detailed description in lines 112-116.

Experimental design

no comment

Validity of the findings

no comment

---

## Round 0.2 · Minor Revisions

The authors should improve the description of the neural network in the revised version of the manuscript.

Reviewer 1 ·

Basic reporting

The authors have improved the clarity and the discussions raised showing sufficient knowledge in the field background.

Experimental design

Authors have further shown satisfactory and detailed discussion of all the methodology steps in the proposed Faster R-CNN.

Validity of the findings

Authors have shown from the comparison table of the experimental results, that the performance of Faster R-CNN framework was effective in the real-time detection of Rice Leaf Diseases when compared with some of the state-of-the-art methods.

Additional comments

Authors have carefully addressed all the issues raised in previous comments.

Reviewer 2 ·

Basic reporting

This is a well-written paper containing interesting results which merit publication. For the benefit of the reader, however, certain statements require further justification. There are given below.
1. In the Model architecture with Faster R-CNN part, more mathematical inference can help readers understand the network.
2. More detail can be shown about the architecture of CNN and ROI in Figure 4.

Experimental design

no comment

Validity of the findings

no comment

Additional comments

no comment

---

## Round 0.3 · accepted · Accept

The article is accepted for publication.